# Evaluation of the Impact Caused by the Snowfall after Storm Filomena on the Arboreal Masses of Madrid

María Eugenia Pérez-González [1,*], José María García-Alvarado [1], María Pilar García-Rodríguez [1] and Raimundo Jiménez-Ballesta [2]

1 Department of Geography, Universidad Complutense of Madrid, 28040 Madrid, Spain; josemaga@ucm.es (J.M.G.-A.); mpgarcia@ucm.es (M.P.G.-R.)
2 Department of Geology and Geochemistry, Universidad Autónoma of Madrid, 28049 Madrid, Spain; raimundo.jimenez@uam.es
* Correspondence: meperez@ucm.es

**Abstract:** Following a copious and famous snowfall in Madrid city (Spain) and its surrounding area, the tree masses were analysed to assess the impact of this snowfall. In this way, this paper proposes an approach towards urban forest impact inventory mapping as a consequence of the snowfall of January 2021, and the subsequent atmospheric stability, which kept surface temperatures in Madrid close to −10 °C. The study has been carried out using snowfall data from 1920 to 2020 and images from the Sentinel, Landsat and MODIS satellites. The results obtained by means of image changes in the NDVI reveal a clear impact on the trees, with 11% of the winter vegetation cover of the municipality being affected. Especially significant has been the damage detected both in the forest areas in the city and in the parks, gardens or urban roads.

**Keywords:** climate risk; snowfall; impact on trees; satellite images; satellite data; NDVI; Sentinel 2; Madrid

## 1. Introduction

Natural disasters occur with some frequency in the western Mediterranean region associated with intense rainfall. Less frequent is the appearance of extensive and intense snowfalls such as that which occurred in January 2021, particularly in Madrid as a consequence of Storm Filomena. Indeed, this storm caused serious disruptions and even substantial damage locally and wreaked havoc over a large part of the Spanish territory, fundamentally as a result of one of the heaviest, most copious snowfalls in decades. This had a particular impact in the city of Madrid and its surrounding area, where the snow cover exceeded 50 to 60 cm, persisting for several days [1].

Handmer et al. [2] point out important scientific advances in relation to the response of precipitation extremes to climate change, probably due to its environmental and human transcendence. However, changes in the appearance of extreme snowfall events have hardly been studied at all, probably because there are few records of snowfall extent and thickness. Indeed, if they exist, they are somewhat regional studies, as Kunkel et al. [3] stated.

Snowfall is a common meteorological phenomenon and a pervasive global hazard. Snowfalls are relatively common, natural and occasional hazards in the Madrid region, where they rarely affect human lives and have limited direct impacts. However, events such as those experienced in January 2021 can sometimes happen. Compared to open-land areas, tree areas intercept snow, which means that snow cover in tree areas tends to be comparatively lower. This fact generates a decrease in the level of evaporation, while there is a greater winter snowmelt under the canopy [4], a process that took special relevance in the case study (that we have been able to verify directly). According to Link and Marks [5,6], variation of forest canopy leads to uneven interception and to small-scale spatial variation in snow accumulation beneath the forest canopy.

Despite the fact that Madrid has a high environmental fragmentation of the city [7], the city of Madrid treasures a green space that in its beginnings acquired an ornamental, recreational and hygienist character (18th to 20th centuries) according to [8,9], at a time in which it opted for resistant species, especially to summer drought. This is the case of massive planting with conifers (mainly Pinus, Cupressus and Cedrus genera), still dominant today. The previous phase has been followed (during the 20th century) by a phase mimicked by the US and Europe, qualified as "green" planning, whose fundamental focus lies on the benefits of these spaces in terms of health, urban ecology, conservation and quality of life [10–14]. A special interest in this planning is its role in buffering the urban heat island [15–23]. More recently, the beneficial role of urban trees has regained strength with respect to both public health [19,24,25] and the effects of climate change [26,27], without neglecting its biological filter action [28,29].

As a result of a long and historic trajectory, the city of Madrid is one of the most wooded cities in the world (with about 5,700,000 units), only behind Toronto, Atlanta, London and Birmingham [30]. These data are historically linked to royal possessions (Monte de El Pardo and Casa de Campo) and to successive municipal actions, city council and private actions, which have culminated until the formation of the numerous copses, parks, gardens and "green" urban street of the city.

In this context of a "green" city, the trees of Madrid have been seriously affected at the beginning of 2021 by the passage of a strong storm, called Filomena by the AEMET, [31], which left heavy snowfall. This unique snowfall has caused significant damage to the trees in the city of Madrid. It is then worth asking what imprint this snow leaves, especially as it is reflected in the falling of numerous trees.

Snowfall constitutes a frequent problem in the world. Snowstorms occur rather rarely in the Mediterranean region, it is therefore worth analysing the long-lasting cold spell and the heavy snowfall associated with Filomena and its effects on forest stands. According to a report issued by AEMET [31], the intense snowfall and widespread snow cover in Spain associated with Filomena were exceptional. However, other events are known (all of which occurred in January) that, although they were not as intense or comparable to the one discussed here: One happened in 1979, another occurred in 1997 and mainly affected mountainous regions in Northern Spain, in particular the Pyrenees. The third strongest snowstorm, in 1986 and also affected mountainous regions. Finally, the most similar and relatively recent is the heavy precipitation event swept over Catalonia (NE Spain) in March 2010, with a total amount that exceeded 100 mm locally and snowfall of more than 60 cm near the coast [32].

Extreme snowfall risks can lead to disasters and fatalities that tend to primarily affect infrastructure and the environment. This is the case of the famous snowfall of January 2021, related to Storm Filomena, which our understanding of the effects remains very limited today. The tree mass of Madrid city covers approximately 26% of the land area of the city [30].

The objective of this study is to inventory and map an estimated forest damage areas linked to the January 2021 snowfall that occurred in Madrid. The visual and digital analysis of different satellite images (Modis, Landsat Sentinel 2) allows us to obtain relevant thematic information (snow cover, surface temperature of the snow cover and variations in vegetation), which offers a first estimate of damage in the trees and allow us to know the sectors most affected by the snowfall. The cartography and analysis of satellite images have been developed using GIS techniques with its ArcGis Pro and Erdas Imagine 2020 software.

## 2. Material and Methods

### 2.1. Study Area

The city of Madrid is in the centre of the Iberian Peninsula and the Central Plateau, ($40°25'00''$ and $3°42'00''$), occupying an extension slightly greater than 600 km$^2$ and a resident population of around 4,500,000 inhabitants (Figure 1). Topographically, it occupies a flat to undulating surface (with small hills and elongated troughs), with an average

altitude of 655 m.a.s.l., reaching the highest point in the north of the city, (740 m.a.s.l.), and the lowest in the southern limit, (554 m.a.s.l.). This territory is in the "Campiña" countryside, a transition area between the Sierra de Guadarrama and the Madrid basin, on tertiary Neogene materials, of a detrital nature in the north and centre (arkoses, sands, clays and silts), and mostly of chemical origin to the south and southeast (marls, gypsum, gypsum marls, including sepiolite and flint) [33]. It is constituted by small hills and elongated troughs, with a dominant NS direction, which other authors have already analysed in their influence on the urban climate of Madrid [34].

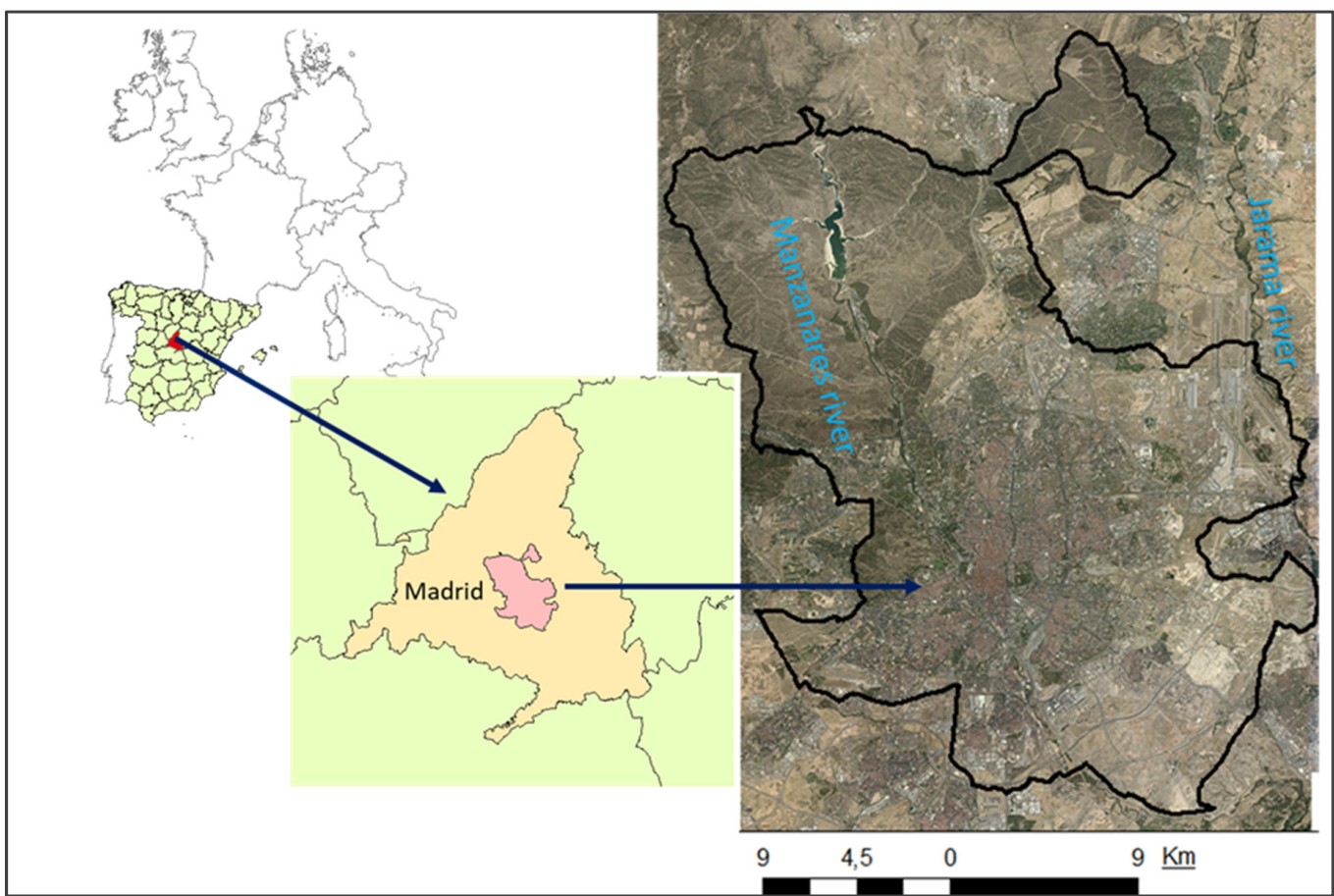

**Figure 1.** Geographical location of the study area.

From a climatic point of view, it is Mediterranean with a continental hue. In the period 1981–2010, the mean annual temperature was 14.5–15 °C and the annual rainfall was 371–428 mm, with a summer drought from June to September and equinoctial maximums, the months of October, November, December and May being the wettest (41–60 mm/month). In addition, the average number of hours of sun per year is estimated to have been between 2600 and 2800 h.

The period with probability of snowfall in Madrid in the last century (1920–2020) occurs from October to April, with a reduced average annual number of days (3.8), but with maximums that reach 11 days/year (1941, 1950, 1978 and 2009) and another 11% of years without snow (Figure 2).

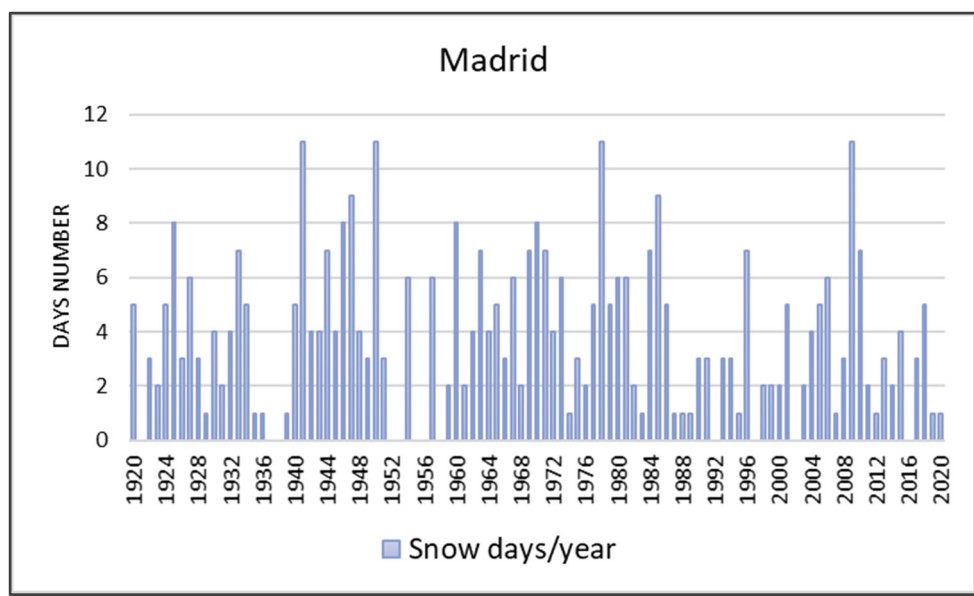

**Figure 2.** Snow days in Madrid during the period: 1920–2020.

The seasonal distribution of snow in the city of Madrid during the last century (1920–2020) shows 80.6% of snow days in the winter season, followed by spring (16%), and to a lesser extent autumn, with 3.4% (Table 1). The month with the highest risk of snow is January (31%) and the one with the lowest risk is October, with 0.26%, since it has only happened once in a century of recorded days (in 1975).

**Table 1.** Percentage distribution of snow days in Madrid, 1920–2020.

| Winter | | | Spring | | Autumn | |
|---|---|---|---|---|---|---|
| 80.57 | | | 16.06 | | 3.37 | |
| December | January | February | March | April | October | November |
| 22.02 | 31.09 | 27.46 | 11.92 | 4.15 | 0.26 | 3.11 |

From the edaphic point of view, the soils of the municipality of Madrid would correspond to different units of Luvisols, Cambisols, Regosols, Arenosols and Calcisols [35]. Gypsisols dominate in the southeast and Fluvisols dominate in the river plains. However, most of the soils in the city have disappeared by sealing or have been transformed into Anthrosols and Technosols.

*2.2. The Tree Mass of Madrid*

The municipality of Madrid, with 5,700,000 trees, has a tree cover of 26%, distributed among forest areas (35% of Monte de El Pardo, Finca de Tres Cantos, plains of the Manzanares and a small part of the Southeast Regional Park), non-municipal conservation green areas (34.4%) and municipal conservation green areas and street trees (30.5%, in Casa de Campo and Parque del Retiro) [36]. Just over half is made up of evergreen species, with a clear predominance of pines (*Pinus pinnea* and *Pinus halepensis*, 34.3%), holm oak (*Quercus ilex*, almost 17%) to which are added *Cupressus arizonica*, *Platanus hybrida*, *Ulmus pumila* and *Sophora japonica* (16.2%) [37,38].

Additionally, a second typology of green spaces must be taken into consideration: small or medium-sized landscaped areas inserted in the urban fabric. Finally, it is important to mention the so-called alignment or road trees, which, according to the municipal inventory, have more than 293,000 specimens of 800 different species.

*2.3. Methodology and Data Pre-Processing Sources Used*

The monitoring of the meteorological situation caused by Storm Filomena has been carried out both from the synoptic maps of the surface and 500 hPa of the German meteorological web server, [37,39] and from the data on snow height recorded by the Spanish Meteorological Agency [31]. The synoptic interpretation is carried out manually, according to the types of weather established by [40], since there is no automatic classification that offers good results in Mediterranean climates.

Regarding the evaluation of the snowy surface and the analysis of the impacts caused on the trees, it is analysed at different scales: a small initial one, to determine the extension of the snow cover in the Iberian Peninsula, even the great one, which has made it possible to estimate its impact on the urban forest of Madrid. Therefore, depending on the work scale, different images have been acquired and processed (Table 2):

- ■ Daily images from the MODIS (Moderate-Resolution Imaging Spectroradiometer) satellite, Terra sensor, with 36 spectral channels, between the visible, near, medium and thermal infrared, with a spatial resolution of 250, 500 and 1000 m, respectively, have been used for the general analysis of the extent of snow cover in the Iberian Peninsula and in particular in Madrid. Here, we have selected the combinations in natural colour (1-4-3, R-V-A) to monitor cloudiness, and in false colour 3-6-7 (R-V-A), to discriminate and measure snow cover. These spectral bands correspond to the blue channel (3: 459–479 μm) and mid-infrared (6: 1628–1652 μm and 7: 2105–2155 μm), with a spatial resolution of 500 m, which when combined with band 1 (visible, at 620–670 μm) an image with a final resolution of 250 m is obtained, offering images with greater detail. The source for consulting and downloading images has been the NASA Worldview server [39–41].

- ■ A Landsat 8 image, with a spatial resolution of 30 m in the visible and near infrared and a thermal channel at 100 m, selected a few days after the passage of the storm (1 November 2021) from the US Earthexplorer server [40,42]. With this, the snow cover in the Community of Madrid was analysed visually and spectrally and the surface temperature of the Community of Madrid and neighbouring areas was calculated, from band 10, thermal infrared with 10.30–11.30 μm, according to the methodologies of Valor and Caselles [43] and Jiménez-Muñoz et al. [44].

- ■ Images from the European Sentinel 2 satellite, with 10 and 20 m spatial resolution in the visible and infrared bands, selected before and after the passage of the storm from the Copernicus download server, processing level 2 [45].

Various spectral bands can be combined with the red-green-blue (RGB) colours used by cartographic and photographic processors. Thereby, bands from the infrared or visible spectrum are combined with RGB colours to obtain false colour images that highlight different land covers [46].

There has never been a satellite mission dedicated to snow mass (SWE) detection. We decided to choose and use the Sentinel 2 data to calculate the impact of snowfall in arboreal masses, because it is a satellite that provides free images every 5 days and with adequate spatial resolution (10 m) [47]. The European Sentinel 2 satellite operates with two twin satellites (S2A and S2B, launched in 2015 and 2017, respectively) and has 13 spectral bands that record the visible, near infrared and mid-infrared [48].

In order to differentiate the spectral responses of the main tree masses damaged by the weight of the snow, the spectral profiles of 4 categories were obtained before and after the snowfall: (*Pinus*, *Quercus ilex*, herbaceous, and bare soil), before and after the passage of Storm Filomena.

In these images, the different land covers of the municipality of Madrid and the variations of the vegetation have been analysed through digital indices and treatments, also suitable for the detection of plant biomass in urban areas [49], which are:

- ✓ Natural false colour band combination (B8-B4-B3, R-V-A).

✓     Supervised classification of the Sentinel 2 image of 6 January 2021, according to the parametric rule of minimum distance.

✓     Normalized Difference Vegetation Index (NDVI): based on mathematical algorithm between parts of the electromagnetic spectrum where vegetation has high reflectance (near infrared) and where vegetation has low reflectance (red) [50] (IRc-red)/(IRc+red).

✓     Images of changes between the NDVI values of February 2020 and those of 2021. These dates have been chosen to collect the differences in the reflectivity of the plant biomass of two upcoming winters, before and after the passage of Storm Filomena.

Together with the sources and methodologies indicated, an approach to the knowledge of urban green spaces in Madrid of a cartographic, documentary, bibliographic and statistical type is also developed: cartographic and statistical sources, focused above all on the Data Bank of the Madrid City Council [51]. With them, a district-based thematic cartography has been elaborated on two essential aspects that allow linking vulnerability and damage in the meteorological phenomenon studied: urban green space and trees.

**Table 2.** Selection of satellite images used.

| Images | Date | Areas Analysed and Affected by Snow | Bands Used | Spatial Resolution (m) |
|---|---|---|---|---|
| MODIS/Terra | Daily of 6 January 2021 to 24 January 2021 | Spanish peninsular centre | 1-4-3 3-6-7 | 250 (Visible) 500 (NIR) 1000 (TIR) |
| Landsat/8 | 12 January 2021 | Comunidad de Madrid (Scene 201/032) | 1 a 7, 10 and 11 | 30 (Vis, NIR and SWIR) 100 m (TIR) |
| Sentinel/2 | 6 January 2021 11 January 2021 18 January 2021 23 February 2020 27 February 2021 | Municipality of Madrid (Scene 30TVK) | 1 a 4 8, 11 and 12 | 10 m (Visible and NIR) 20 m (SWIR) |
| Photos PNOA | 2016 | Leaves 533, 534, 559 y 582 | Natural Color | 50 cm |

Sources: NASA (2021) [39,41], USGS (2021) [40,42], Copernicus Sentinel Hub (2021) [45,48], and IGN (2021) [52].

Finally, intense field work has been carried out, which has allowed the compilation of numerous photographs of the damage, especially in the trees, and contrasting the ground truth with the results obtained.

## 3. Results and Discussion

### 3.1. Storm Filomena, Features and Extension

Reviewing the official reports from the Agencia Estatal de Meteorología, as a first approximation, meteorologically the intense snowfall is produced by the conjunction and succession of several episodes. Tapiador et al. [1] detail the meteorological events that surrounded it. After the entry of very cold arctic air, between 31 December 2020 and 6 January 2021, the formation and deepening of an Atlantic storm occurred between Madeira (Portugal) and the Gulf of Cádiz (South of Spain), which left a strong storm of rain and wind in the Canary Islands and the south of the peninsula (specifically on 6 and 7 January 2021). The storm spread towards the centre and northeast of the Iberian Peninsula on the 8th and 9th, leaving large amounts of snow, and a large blanket not remembered in many decades, so it is the most extensive snowfall in recent decades, possibly to be classified as historical.

Once the storm was removed on the 10th, it was followed by an anticyclonic situation, with a cold wave and extremely low thermal values, which allowed the snow cover to be maintained for about ten days, until another storm entered from the W, on the 19 January 2021.

The first consequence of Storm Filomena was the extraordinary snowfall, both in extent and in thickness. The general approximation of the snow extent reached in the Iberian

Peninsula after the passage of Storm Filomena can be followed through the MODIS/Terra satellite images (Figure 3). The visual interpretation in false colours is very expressive, especially when combining the visible and mid-infrared channels. Of the many possible false colours, the 3-6-7 (R-V-A) band combination easily discriminates snow in bright red, easily differentiating it from cloudiness in white-orange tones and from the snow-free surface in green.

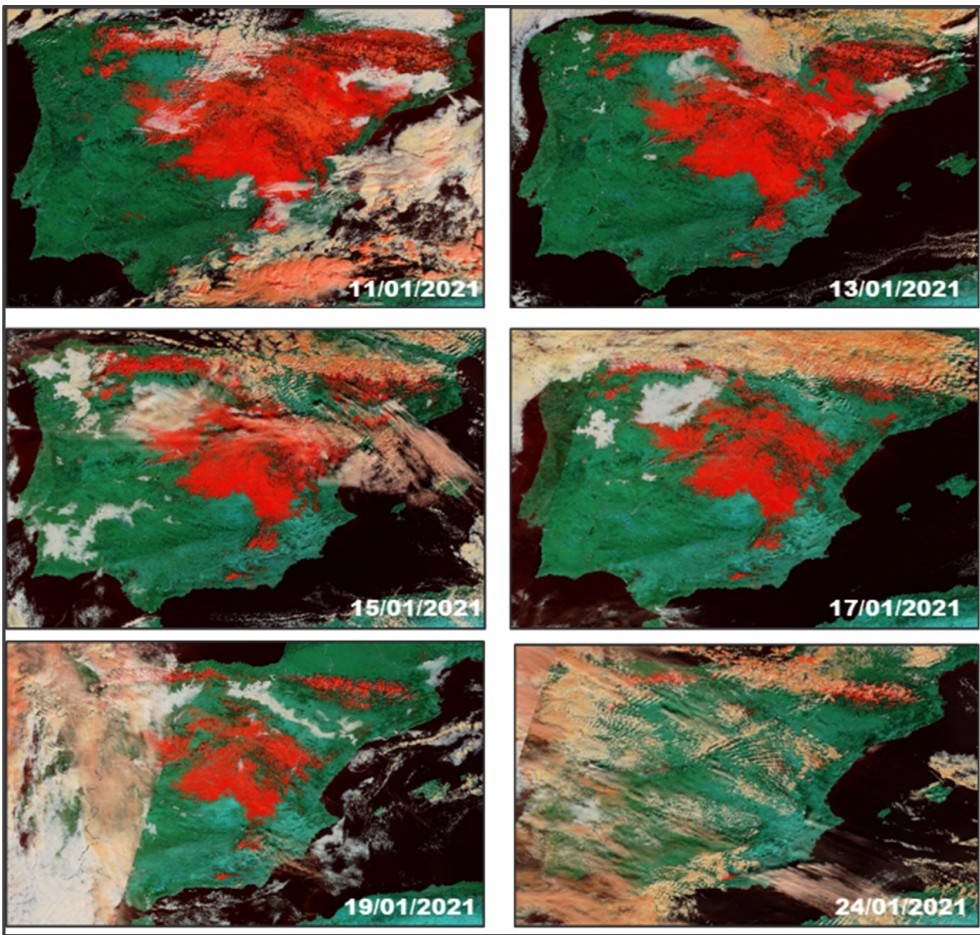

**Figure 3.** MODIS/TERRA images, combination 3-6-7 (R-V-A). Source NASA (2021) [39].

In greater detail, the Landsat image of 1 November 2021 has made it possible to identify the covered and free sectors of the snow cover not only in the city of Madrid but throughout the Community of Madrid.

In the municipality of Madrid, temperatures ranged from the minimum, −11 to −10 °C, in forest areas and bare soils (cyan tones), to the least low occurring in the central district, with −3.5 to −5 °C (pink tones and reddish) (Figure 4). Despite the exceptionally cold values reached on those days, the thermal pattern of the "heat island" of Madrid remains, meaning that the less rigorous values appear in the centre; by contrast, the lowest ones appeared in the forest spaces and in the highest districts of the city [53]. These lower temperatures of the urban green enclaves [15,21,54] exceptionally low on these dates, will have a clear negative impact on the forest of the municipality of Madrid.

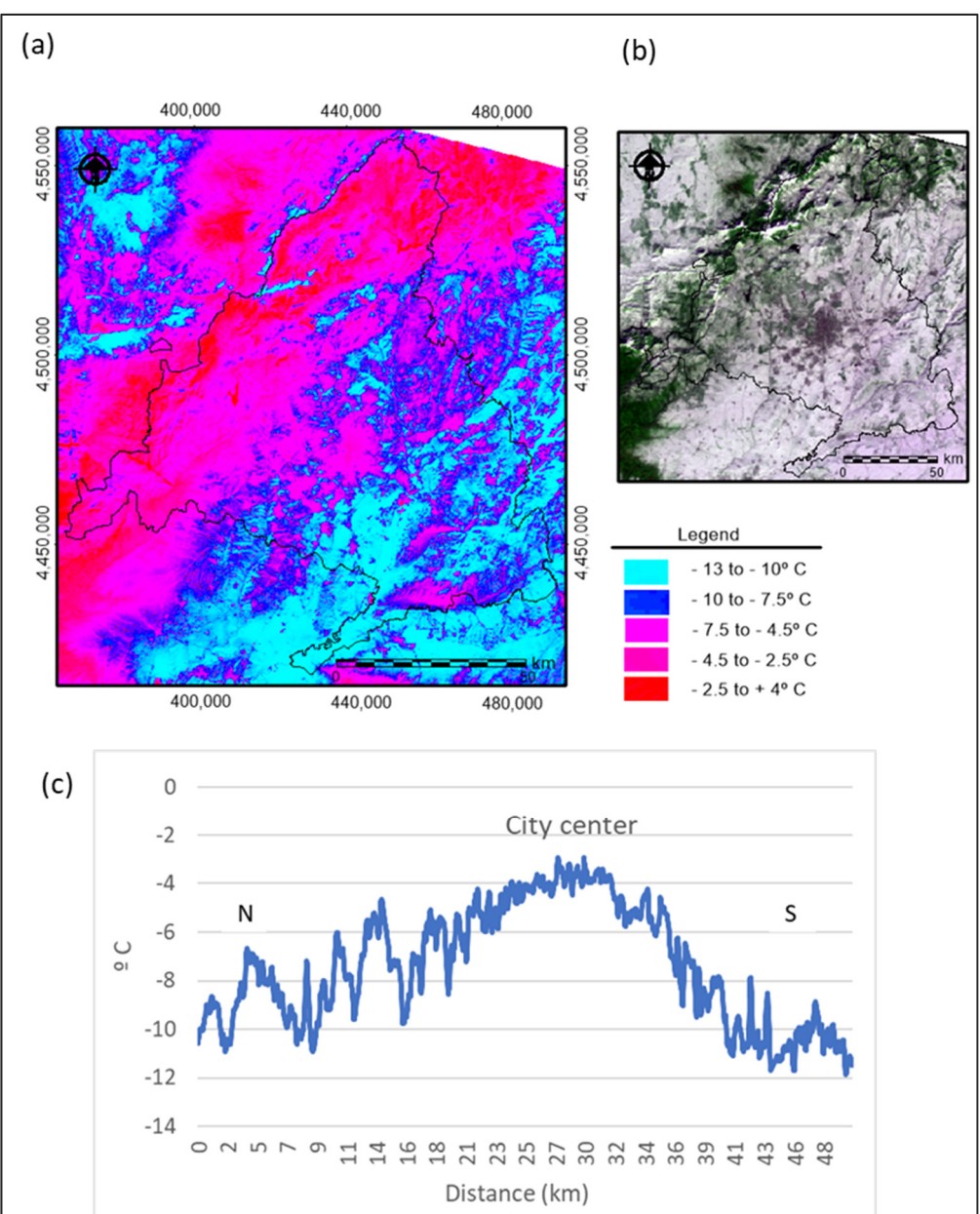

**Figure 4.** Landsat 8 image, 1 November 2021: surface temperature (**a**) natural false colour image (**b**,**c**) surface temperature profile. Source USGS (2021) [40].

*3.2. Estimation of Damage to the Tree Mass Using Satellite Images*

The analysis of the severe convective storm of January 2021 in Madrid, using different sources of observational data, helped us to understand the damage of woodland. With a plant cover spread of over 11,356 ha (Figure 5), the municipal green count of Madrid amounts to 25.43%, a figure in line with that provided by Morcillo San Juan et al. [36].

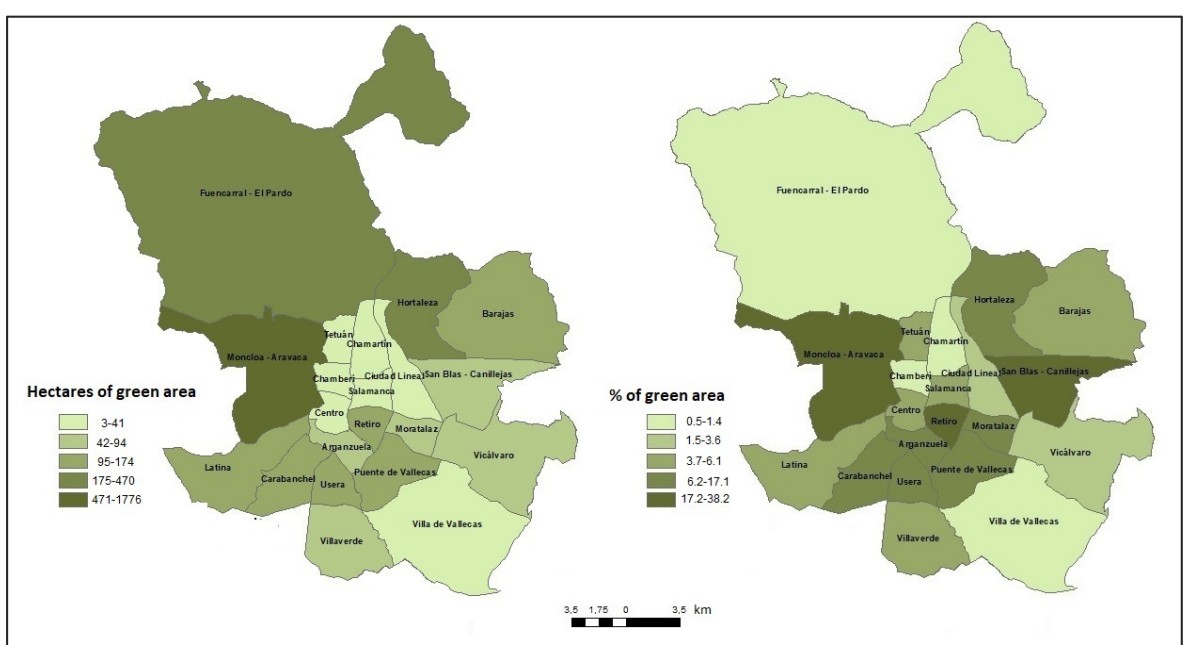

**Figure 5.** Absolute and relative green space in the districts of the municipality of Madrid. Source Ayuntamiento de Madrid [51].

It should be noted that the percentage values of the forest surface of Monte de El Pardo are abnormally low due to the predominance of a sparse oak forest, "dehesa", with a large part of bare and/or herbaceous soil (Figure 5).

Despite the fact that all the green space suffered damage from the heavy snowfall (and the subsequent persistence of several days of snow accumulation), it is the trees that initially seem to have been the most attacked (Figure 6).

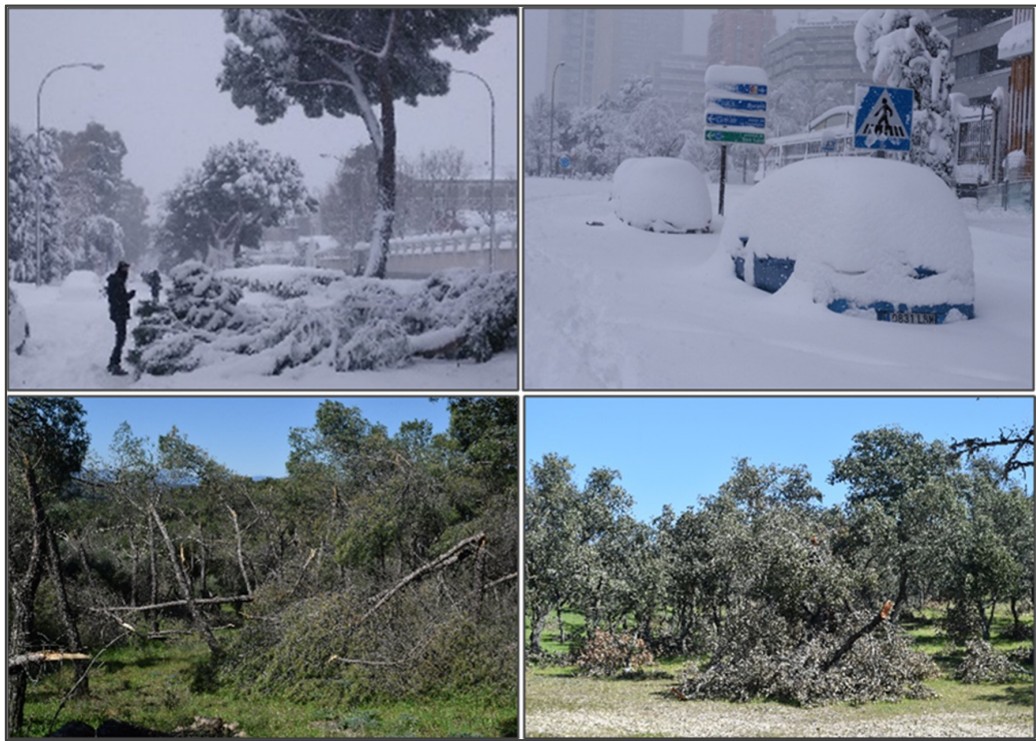

**Figure 6.** Snow accumulated by the Filomena Storm and impacts on trees: examples on urban roads and urban parks.

However, given the high recurrence of Sentinel 2 images (every 5 days), this allows us to determine the extent of the vegetation cover prior to the snowfall (1 June 2021) and the areas affected by the snow (Figure 7).

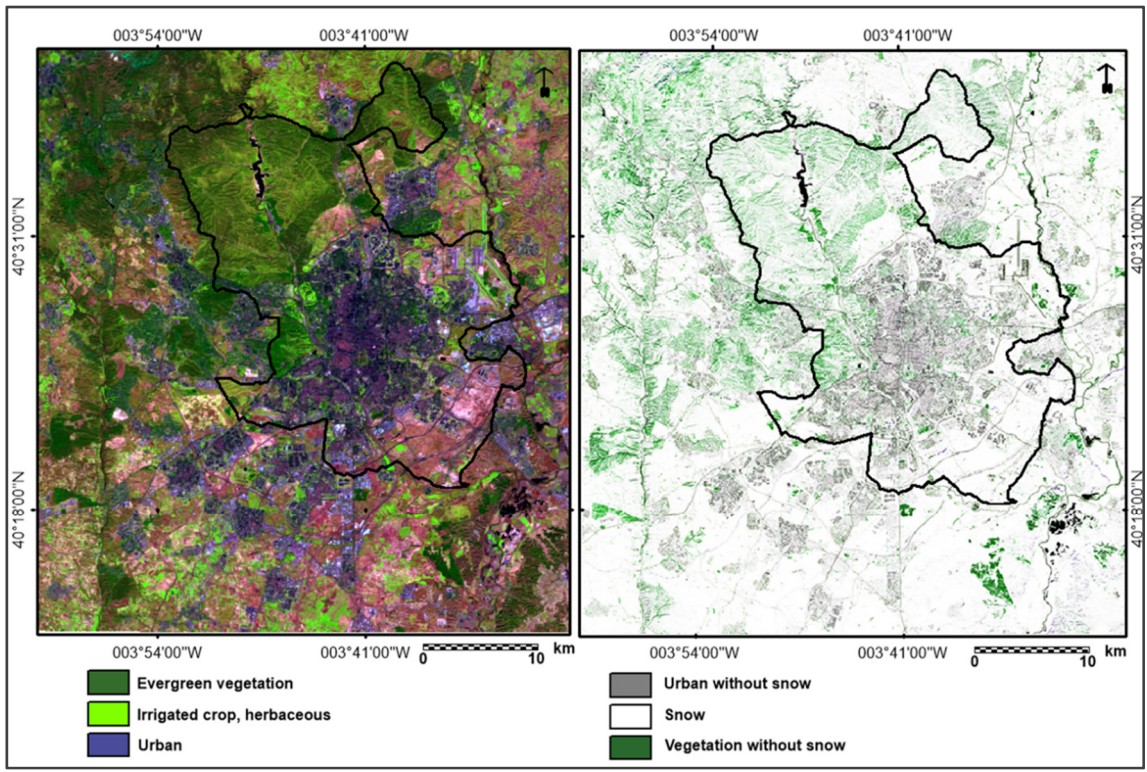

**Figure 7.** Sentinel 2 images. Left: Infrared false colour from 1 June 2021 before the snowfall; right: from 1 November 2021, after the same. Source Copernicus Sentinel Hub [45,48].

In the false-colour image of the first date, the surface of evergreen vegetation (mainly *Pinus pinea, Quercus ilex* and *Pinus halepensis*, together with its associated shrub fringe) is represented in green, more or less dark depending on the density; therophyte herbaceous, lawns (golf courses and private) and some irrigation in light green; the urban fabric and communication routes in bluish and violet tones; and, finally, the bare soils in light tones (whitish and pink).

In the image of 1 November 2021, the surface of the Madrid metropolitan area is divided into three categories: snow cover (white), buildings and road infrastructure (grey tones) and snow-free vegetation (green tones). In this image it can be seen that part of the wooded area is free of snow, since initially it is retained in some branches, which in many cases do not support the additional weight and end up falling, sometimes dragging the entire tree. However, the great extension of the snow cover is visible throughout the municipality of Madrid, which affected both the forest areas and the peri-urban fringe, with the lowest surface temperatures ($-11$ to $-7.5$ °C), as well as the interior of the city since it also remained below zero for more than a week (Figures 4 and 7).

Undoubtedly, the great coverage of snow on the various green spaces, and its persistence in the form of frost must have had different consequences for the elements that compose them (deciduous trees, evergreen trees, non-native gardening, grasslands, etc.). A differentiating element of this impact lies undoubtedly in the physiology and morphology of the plants they are composed of in relation to the meteorological picture experienced in the aforementioned days, in their adaptation or non-adaptation to the natural environment and the rarity of this meteorological event in our environment.

Figure 8 shows the high contrast in the values of all the spectral signatures, before and after the Storm Filomena. The decrease in reflectance values is due to the loss of plant mass

and the increase in soil moisture. However, the profiles corresponding to the vegetation maintain in both cases the highest values in the near infrared (band 8). Likewise, the profile of the bare soil maintains the maximum reflectance in the mid-infrared (bands 11 and 12).

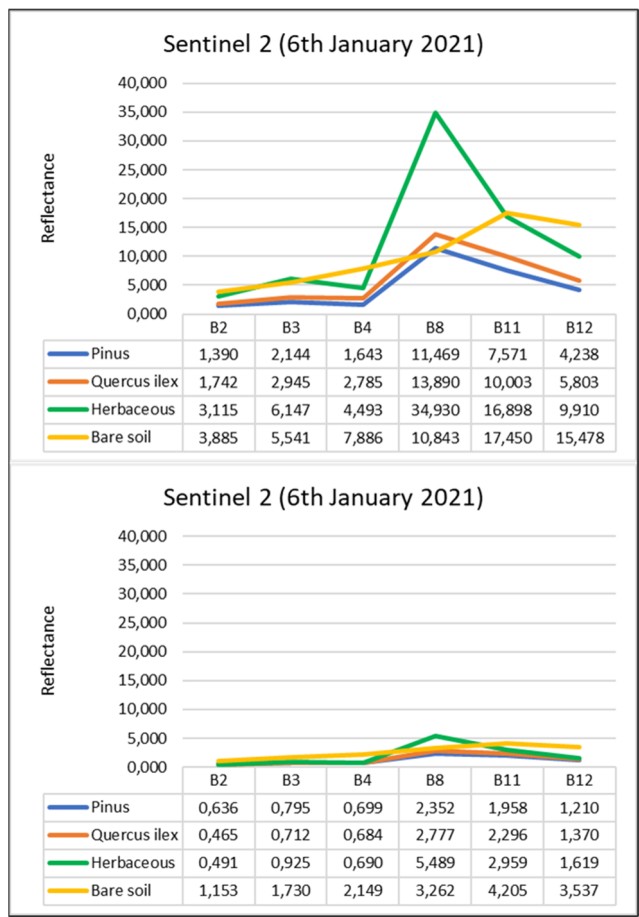

**Figure 8.** Differences in the spectral signatures (before and after the Storm Filomena) obtained from the bands of the Sentinel 2 satellite.

In principle, the most visible physical impact has been the fall of branches and tree trunks, fundamentally due to the weight of snow accumulation and its subsequent freezing, which resulted in sustaining the pressure on them for longer. To this direct reduction of the trees must be added the intense logging and subsequent forced fellings, given the imminent risk to the population or to the viability of the trees themselves. These tasks continued in the city for several months, working first on the street trees, to later undertake the numerous urban parks and forest areas. In the larger spaces (Monte de El Pardo and Casa de Campo), the volume of material to be removed was such that, after four months of the event, there was still much evidence of the damage caused, which will take years to recover.

Among the tree species that make up the copses, parks, gardens and are part of the Madrid street map, those most affected by the support of the snow have been the conifers. Among them, the most affected have been the pine forests, also the most abundant (*Pinus pinea* and *Pinus halepensis*), with a notable loss of trees and branches in some neighbourhoods of the city, and *Cuppresus* sp., more due to the doubling of their branches than by falling individual trees. To a lesser extent, firs and cedars have also been affected, since their pyramidal physiognomy and more open arrangement of branches give them a better adaptation to the weight of the snow. In some parks and private gardens, the losses due to the freezing of some non-native and exotic species (*Arecaceae*, *Agavaceae* or *Cactaceae* families) therein are not negligible, but quantitatively they are very minor.

In Monte de El Pardo, the loss is notable in the holm oak forests, adding to past damage for phytosanitary reasons, which caused a high drought [55]. However, on this mountain the impact is more intense in the repopulation sectors of *Pinus pinea*, with a massive fall in specimens and thousands of branches (Figures 6 and 9).

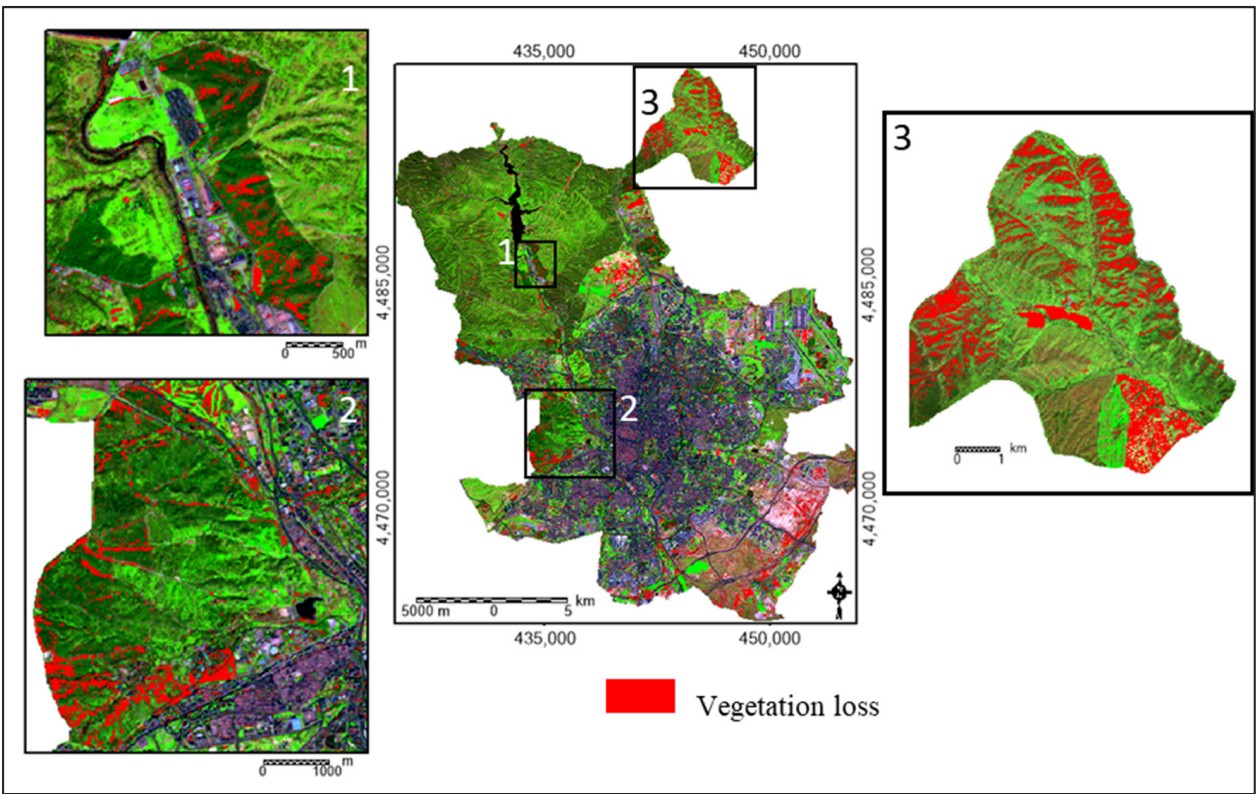

**Figure 9.** Loss of vegetation in the municipality of Madrid and details in different sectors, obtained through the image of changes in the NDVI between February 2020 and 2021. 1: Repopulation pine forests of Monte de El Pardo. 2: Repopulation pine forests of Casa de Campo; 3. Holm oak and pine forests to the north of the municipality. Sources Copernicus Sentinel Hub [48].

The estimation of the loss of plant biomass, after the passage of Storm Filomena in Madrid, obtained from the images of changes in the vegetation index (NDVI), shows the intense impact on the entire municipality (Figure 9). The loss of trees in the urban road logically affected the neighbourhoods with more conifers, predominant to the E and NE of the municipality (districts of Ciudad Lineal and Hortaleza), but also to the pine forests of the margins of one of the main communication arteries, the M-30, which, growing on a slope and in shallow soils, have fallen in large numbers. To the SE and S of the municipality, the notable losses of vegetation affected therophytic herbaceous plants, which recovered better, but also conifers from the Pinar de Santa Eugenia and Entrevías Forest Park. However, there are sectors that have been particularly punished, among which some examples should be highlighted (Figure 9):

- Repopulation pine forests of Monte de El Pardo and Casa de Campo, with high density (1 and 2, Figure 9).
- Holm oak and pine forests to the north of the municipality (Finca de tres Cantos—Monte de Viñuelas), in the shady exposures, since they are the ones that harboured icy temperatures for the longest time (3, Figure 9).

In these examples, and in general in all the parks with a predominance of conifers, the damage is very intense where the support of the trees was more compromised, because they are on small slopes, or next to paths, with highly degraded soils (Arenosols and mainly

Regosols). This fact is represented by very narrow and rectilinear red sections in the NDVI change image (Figure 9).

The calculations and images obtained from the NDVI of the medium resolution multispectral images offer information on the vegetation response, very precise in its spatial location.

In quantitative terms, the municipality of Madrid lost around 11% of its winter vegetation cover, rising to 14% in the districts with the two great lungs of the city: El Monte de El Pardo and Casa de Campo. However, some sectors of the northeast of the municipality, with more rugged orography and higher altitude, reach higher damages in the forest mass.

Other tree species with less spatial representativeness and poor adaptation to the cold, such as palm trees, have been severely damaged by this snowstorm. However, they are only representative in some historical parks, such as the Madrid Botanical Garden, where most have disappeared. Additionally, there really are no citrus fruits in Madrid as an arboreal element (frequent in the east and south of Spain).

The notable loss of evergreen trees after the passage of Filomena left Madrid less protected from the frequent episodes of winter pollution, and with greater lighting in some streets and urban green spaces, which was to predictably lead to increases in temperature.

The connection of these types of extraordinary events with global warming is still uncertain. Nevertheless, [56] introduces a theory for the response of snowfall extremes to climate change. The global mean temperature is expected to rise 1.0–5.7 °C by the end of the twenty-first century [57,58], so these types of atmospheric situations are expected to take place each time with greater profusion [53,57]. Therefore, it is necessary to take into consideration both management measures by the relevant authorities and an improvement in the study of satellite images for these cases. In this way, although the combined use of satellite images of different spatial resolutions (MODIS/TERRA, Landsat 8 and Sentinel 2) has made it possible to better understand the extent of this extraordinary snowfall and to estimate some of the impacts on evergreen trees, in order to assess the real losses, satellite images, photogrammetric flights or current and more detailed lidar images are required; this aspect will be convenient to carry out when the tasks of felling and removing the damaged tree material are completed. For this, it would be necessary to use images of greater detail, which allow a more precise classification [49], or lidar images, which discriminate the vegetation by heights [36], an aspect that can be dealt with in future investigations. What would be ideal to do is to determine the effects of snowfall under future climate scenarios, using publicly available snowpack projections under future climate scenarios. In the bibliography, there are climate models, such as the CMIP5 [59], which were applied for historical and representative concentration pathway conditions.

## 4. Conclusions

This paper evaluates and quantifies the preliminary impact of heavy snow on the Madrid municipality. The main results are:

- The height of snow reached in the city of Madrid (52.9 mm) is unprecedented in the last century, so it is considered rare and extraordinary, although it has happened in the month with the greatest risk of snowfall (January).
- The permanence of the snow cover in Madrid for about ten days responds both to the amount fallen and to the subsequent atmospheric stability. Consequently, an intense thermal inversion took place, which left a large part of the municipality of Madrid with frigid surface temperatures (from −13 °C to −2.5 °C), and less rigorous ones in the Sierra (from −2.5 °C to 4 °C).
- The most affected trees have been the conifers because, by keeping their leaves, they retain more snow. Among them, the most affected have been the pine forests, also the most abundant ones (*Pinus pinea* and *Pinus halepensis*), with a notable loss of stems and branches, and cypresses (*Cuppresus* sp.), more due to the bending of their branches than to the fall of individual trees. To a lesser extent, some firs and cedars have also been affected, since their pyramidal appearance and more open branches make them

better adapted to the weight of the snow. In some parks, private gardens and nurseries, the losses due to freezing of some non-native and exotic species in Madrid (*Arecaceae*, *Agavaceae* or *Cactaceae* families) are not negligible, but quantitatively they are very minor. In Monte de El Pardo and Casa de Campo, the loss is also considerable for the holm oaks (*Quercus ilex*).

- In general, in all the parks with a predominance of conifers, regardless of their location, the damage has been very intense wherever the support of the trees was more compromised because they are on slopes, or next to paths and paths with highly degraded soils (mainly regosols).
- Although it is difficult to calculate the forest mass damaged by the accumulation of snow, a first estimate in the municipality of Madrid shows a loss of around 11% of the plant cover in winter, rising to 14% in the districts with the two large lungs of the city: El Monte de El Pardo and Casa de Campo. In both, the damage is maximum in the pine forest repopulation sectors, with high density.
- The combined use of satellite images of different spatial resolutions (MODIS/TERRA, Landsat 8 and Sentinel 2) has made it possible to better understand the extent of this extraordinary snowfall and to estimate some of the impacts on evergreen trees. However, to assess the real losses, satellite images, photogrammetric flights or current and more detailed lidar images are required, which will be convenient to carry out when the tasks of felling and removing the damaged tree material are completed.
- As an essential methodological problem, it should be mentioned that there is enormous difficulty in knowing and accurately mapping the number of trees prior to the storm in Madrid, as the different available sources disagree with each other, since in Madrid numerous juxtaposed ownerships converge and there is a consequent variety of management, production and distribution of information, which makes it especially complex to collect all the information on all green spaces.

The study of extreme events is a topic of inquiry that is becoming more common every day. In this sense, this work transcends the local character and is potentially applicable elsewhere as a basic contribution when it comes to urban forest impact assessments following anomalous snowfall (of large dimensions) in an urban setting.

**Author Contributions:** M.E.P.-G., J.M.G.-A., M.P.G.-R. and R.J.-B. have made substantial, direct and intellectual contributions to the work and approved it for publication. All authors have read and agreed to the published version of the manuscript.

**Funding:** This research has been funded by BANCO SANTANDER—UCM, PR108/20-24: "Update of the susceptibility and risk of flooding in sealed areas of the Community of Madrid and surrounding areas: case studies and proposals for improvement."

**Institutional Review Board Statement:** Not applicable.

**Informed Consent Statement:** Not applicable.

**Data Availability Statement:** Not applicable.

**Acknowledgments:** The authors extend great gratitude to the anonymous reviewers for their helpful review and critical comments. Additionally, we thank the linguistic services at the university for reviewing the written English in the manuscript.

**Conflicts of Interest:** The authors declare no conflict of interest.

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
