# Peer review of "Evaluation of the Impact Caused by the Snowfall after Storm Filomena on the Arboreal Masses of Madrid"

_land, doi:10.3390/land11050667_

Round 1
Reviewer 1 Report
Dear Authors,
I must say that I am biased when snow is concerned so I was eager to read this paper.
Since my primary scientific focus are ITC and its usage in WSN, please take my notifications only as suggestions how to increase overall quality of the paper and user experience while reading it.
1. this paper analyze heavy snow impact on Madrid occurred in 2021, described as one the most heaviest until now:
"This is the case of the famous snowfall of January 2021, related to Storm Filomena, which our understanding of the effects very limited today. Nevertheless, our understanding of the effects of snowfall on Madrid is very limited."
Are there any documented events in surrounding area similar to the event?
Are there any cases of hard snow related to the Storm Filomena?
Could you expand your research, so that we might shed some light to the: "very limited effects of Filomena today".
2. methodology used in this research is combined data from "the synoptic maps of the surface and 500 hPa of the German meteorological web 148 server, and from the data on snow height recorded by the Spanish Meteorological Agency"
Reading the paper one can understand results derived from analyzed data and its significance. Could you detail more about 'how this data are produces (at the moment reader only know where such data arrived from). Please explain which methods and analyzes are used to produce such data. This way, next reader can continue you research and get the best of it in future analysis.
3. This paper, as stated, evaluates preliminary impact of heavy snow on Madrid.
a) If scope of this paper is not how to better understand analyzes of heavy snow impact, but to show how it affected Madrid, than please document more detail what actions could be done to make possible future events 'lighter' on impact area.
b) If this paper aims to help better understanding of similar events, related to the storm Filomena then please document more detail what are suggestions how to deal with.
I believe that this paper has potential and could answer more questions, so I would suggest moderate revision to it.
Thank you for writing it and hope that this suggestions could help you make it even better.
Regards
Reviewer 2 Report
Major critique:
I would argue that the paper’s contribution is an urban forest impact assessment following anomalous snowfall in an urban setting. It would add to the contribution to point out the novelty of snowfall as an extreme event topic of inquiry. This should be highlighted in the abstract, introduction, and conclusion. It is this which transcends the local example and is potentially applicable elsewhere. Similarly, a discussion of the extent to which it is novel the use of the methods chosen to address this issue would be an important contribution to highlight and expand upon.
There are some awkward phrases that could benefit from editing from a native English speaker. For example, just from the introduction:
Title: change ‘’in’’ to ‘’on’’
Line
30: change ‘’consequence’’ to ‘’a consequence’’
35 delete ‘’acting and’’
48 add ‘’a’’ before ‘’process’’
66-69 isn’t a sentence and I can’t figure out the meaning
76 remove ‘’to’’ before ‘’infrastructure’’
78 add a verb such as ‘’are’’ or ‘’remain’’
81 clarify the meaning of ‘’ a first potential forest damage 81 areas’’
Other issues
137 what does ‘’between old’’ mean?
207 I would recommend expanding the discussion of the field work. Currently it is only one sentence and yet it seems to have been important to the methods.
243 delete second ‘’and’’
Figure 5 percent green and figure 7 showing evergreen vegetation cover in the north of the city don’t correspond. Figure 5 suggests low percentage vegetation cover while figure 7 shows very high green vegetation cover. Please reconcile.
Figure 9 – describe the numbers in the caption.
The region depicted in #3 in figure 9 looks to greatly exceed the purported maximum of 14% tree loss. Please explain.
Lastly, it would be interesting to the readership the extent to which exotics such as palm trees and citrus were damaged or destroyed.
Buena suerte!
Round 2
Reviewer 2 Report
The revisions are acceptable. One other issue. What is meant by ''feet'' here? (sq. feet?):
As a result of a long and historic trajectory, the city of Madrid is one of the most 64
wooded cities in the world (with about 5,700,000 feet)
Felicitaciones!
Author Response
Thank you very much for the kind review. It does not refer to square feet, it refers to units (tree).
This manuscript is a resubmission of an earlier submission. The following is a list of the peer review reports and author responses from that submission.